# Learning Dissipative Dynamics in Chaotic Systems

**Zongyi Li**[*]
Caltech
zongyili@caltech.edu

**Miguel Liu-Schiaffini**[*]
Caltech
mliuschi@caltech.edu

**Nikola Kovachki**
NVIDIA
nkovachki@nvidia.com

**Burigede Liu**
University of Cambridge
bl377@eng.cam.ac.uk

**Kamyar Azizzadenesheli**
NVIDIA
kamyara@nvidia.com

**Kaushik Bhattacharya**
Caltech
bhatta@caltech.edu

**Andrew Stuart**
Caltech
astuart@caltech.edu

**Anima Anandkumar**
Caltech
anima@caltech.edu

## Abstract

Chaotic systems are notoriously challenging to predict because of their sensitivity to perturbations and errors due to time stepping. Despite this unpredictable behavior, for many dissipative systems the statistics of the long term trajectories are governed by an invariant measure supported on a set, known as the global attractor; for many problems this set is finite dimensional, even if the state space is infinite dimensional. For Markovian systems, the statistical properties of long-term trajectories are uniquely determined by the solution operator that maps the evolution of the system over arbitrary positive time increments. In this work, we propose a machine learning framework to learn the underlying solution operator for dissipative chaotic systems, showing that the resulting learned operator accurately captures short-time trajectories and long-time statistical behavior. Using this framework, we are able to predict various statistics of the invariant measure for the turbulent Kolmogorov Flow dynamics with Reynolds numbers up to 5000.

## 1 Introduction

**Machine learning methods for chaotic systems.**   Chaotic systems are characterized by strong instabilities. Small perturbations to the system, such as the initialization, lead to errors which accumulate during time-evolution, due to positive Lyapunov exponents. Such instability makes chaotic systems challenging, both for mathematical analysis and numerical simulation. Because of the intrinsic instability, it is infeasible for any method to capture the exact trajectory of a chaotic system for long periods. Therefore, prior works mainly focus on fitting recurrent neural networks (RNN) on extremely short trajectories or learning a step-wise projection from a randomly generated evolution using reservoir computing (RC) [1–5]. Another approach to this problem is to work in the space of probability densities propagated by, or invariant under, the dynamics [6, 7], the Frobenius-Perron approach; a second, related, approach is to consider the adjoint of this picture and study the propagation of functionals on the state space of the system, the Koopman operator approach [8, 9]. However, even when the state space is finite dimensional, the Frobenius-Perron and Koopman approaches require approximation of a linear operator on a function space; when the state space is infinite-dimensional approximation of this operator is particularly challenging. These previous

---

[*]Equal contribution

The code is available at https://github.com/neural-operator/markov_neural_operator

attempts are able to push the limits of faithful prediction to moderate periods on low dimensional ordinary differential equations (ODEs), e.g. the Lorenz-63 system, or on one-dimensional partial differential equations (PDEs), e.g. the Kuramoto-Sivashinsky (KS) equation. However, they are less effective at modeling more complicated turbulent systems such as the Navier-Stokes equation (NS), especially over long time periods. Indeed, because of positive Lyapunov exponents, we cannot expect predictions of long-time trajectories of such chaotic systems to be successful. Instead, we take a new perspective: we capture statistical properties of long trajectories by accurately learning the solution operator, mapping initial condition to solution at a short time later.

**Invariants in chaos.** Despite their instability, many chaotic systems exhibit certain reproducible statistical properties, such as the auto-correlation and, for PDEs, the energy spectrum. Such properties remain the same for different realizations of the initial condition [10]. This is provably the case for the Lorenz-63 model [11, 12] and empirically holds for many dissipative PDEs, such as the KS equation and the two-dimensional Navier-Stokes equation (Kolmogorov flows) [13]. Dissipativity is a physical property of many natural systems. Dissipativity may be encoded mathematically as the existence of a compact set into which all bounded sets of initial conditions evolve in a finite time, and therafter remain inside; the **global attractor** captures all possible long time behaviour of the system by mapping this compact set forward in time [13, 14]. There is strong empirical evidence that many dissipative systems are ergodic, i.e., there exists an invariant measure which is supported on the global attractor. For autonomous systems, our focus here, the **solution operator** for the dynamical system maps the initial condition to the solution at any positive time. By fixing an arbitrary positive time and composing this map with itself many times, predictions can be made far into the future. This compositional property follows from the Markov semigroup property of the solution operator [13, 14]. It is natural to ask that approximations of the solution operator inherit dissipativity of the true solution operator, and that invariant sets and the invariant measure, which characterize the global attractor, are well-approximated; this problem is well-studied in classical numerical analysis [15, 16] and here we initiate the study of inheriting dissipativity for machine-learned solution operators. By learning a solution operator, we are able to quickly and accurately generate an approximate attractor and estimate its invariant measure for a variety of chaotic systems that are of interest to the physics and applied mathematics communities [17–23]. Prior work in learning complex dynamics from data [24–27] has labeled long-term stability a desirable feature (e.g., [28, 29] learning the attractors of dynamical systems), but all lack a principled data-driven modeling paradigm to enforce it. In this paper we work within a precise mathematical setting encompassing a wide variety of practical problems [13], and we demonstrate two principled approaches to enforce dissipativity. One of these leads to provably dissipative models, something not previously achieved in the literature for our definition of dissipativity. [2]

**Neural operators.** To learn the solution operators for PDEs, one must model the time-evolution of functions in infinite-dimensional function spaces. This is especially challenging when we need to generate long trajectories since even a small error accumulates over multiple compositions of the learned operator, potentially causing an exponential blow-up or a collapse due to the space's high dimensionality. Because we study the evolution of functions in time, we propose to use neural operators [30, 31], a recently developed operator learning method. Neural operators are deep learning models that are maps between function spaces. Neural operators generalize conventional neural networks which are maps between finite-dimensional spaces and subsume neural networks when limited to fixed grids. The input functions to neural operators can be represented in any discretization, and the output functions can be evaluated at any point in the domain. Neural operator remedies the mesh-dependent nature of finite-dimensional neural network models such as RNNs, CNNs, and RC. Neural operators are guaranteed to universally approximate any operator in a mesh independent manner [31], and hence, can capture the solution operator of chaotic systems. This approximation guarantee and the absorption of trajectories by the global attractor makes it possible to accurately follow it over long time horizons, allowing access to the invariant measure of chaotic systems.

**Our contributions.** In this work, we formulate a machine learning framework for chaotic systems exploiting their dissipativity and Markovian properties. We propose the Markov neural operator (MNO) and train it given only one-step evolution data from a chaotic system. By composing the

---

[2]There are other, more restrictive, definitions of dissipativity; we work with the widely adopted definition from [13, 14].

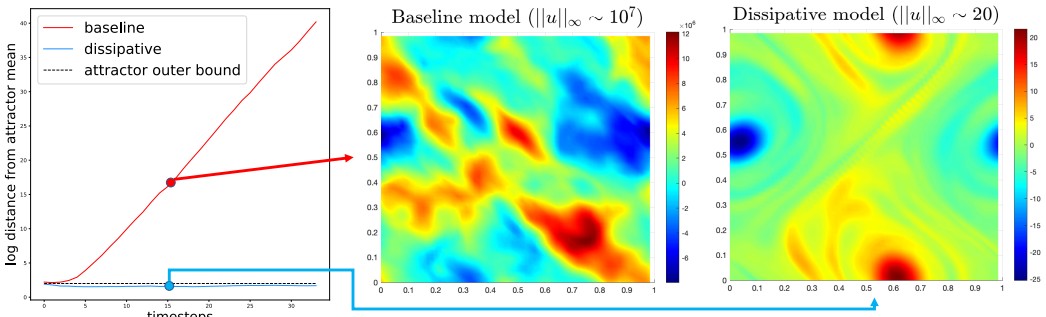

Figure 1: Dynamic evolution of the Markov neural operator for Kolmogorov flow systems Starting from initial conditions near the attractor, with and without dissipativity regularization. The baseline model has no dissipativity regularization while the dissipative model has the regularization enforced during training. Baseline model blows up, whereas the dissipative model returns to the attractor. The dissipative model is trained using the Fourier neural operator architecture in the manner shown in Figure 2.

learned operator over a long horizon, we accurately approximate the global attractor of the system [32]. Our architecture is outlined in Figure 2. In order to assess its performance, we study the statistics of the associated invariant measure such as the Fourier spectrum, the spectrum of the proper orthogonal decomposition (POD), the point-wise distribution, the auto-correlation, and other domain-specific statistics such as the turbulence kinetic energy and the dissipation rate. Furthermore we study the behavior of our leaned operator over long horizons and ensure that it does not blow up or collapse but rather accurately follows the global attractor. In this work:

- We theoretically prove that, under suitable conditions, the MNO can approximate the underlying solution operator of chaotic PDEs, while conventional neural networks lack such strong guarantees.
- We impose dissipativity by augmenting the data on an outer shell to enforce that the dynamic evolution stays close to the attractor. As an additional safeguard, we impose a hard dissipativity constraint on MNO predictions far from the attractor and the data augmentation shell. We show this is crucial for learning in a chaotic regime as demonstrated by Figure 1. The resulting system remains stable against large perturbations.
- We study the choice of time steps for training the MNO, demonstrating that the error follows a valley-shaped phenomenon (cf. [33–35]). This gives rise to a recipe for choosing the optimal time step for accurate learning.
- We show that standard mean square error (MSE) type losses for training are not adequate, and the models often fail to capture the higher frequency information induced from the derivatives. We investigate various Sobolev losses in operator learning. We show that using Sobolev norms for training captures higher-order derivatives and moments, as well as high frequency details [36, 37]. This is similar in spirit to pre-multiplying the spectrum by the wavenumber, an approach commonly used in the study of fluid systems [38].
- We investigate multiple existing deep learning architectures, including U-Net [39], long short-term memory convolution neural networks (LSTM-CNN) [40], and gated recurrent units (GRU) [41], in place of the neural operator to learn the solution operator. We show MNO provides an order of magnitude lower error on all loss functions studied. Furthermore, we show that MNO outperforms the aforementioned neural network architectures on all statistics studied.

In summary, we propose a principled approach to the learning dissipative chaotic systems. Dissipativity is enforced through data augmentation and through post-processing of the learned model. The methodology allows for treatment of infinite dimensional dynamical systems (PDEs) and leads to methods with short term trajectory accuracy and desirable long-term statistical properties.

## 2 Problem setting

We consider potentially infinite dimensional dynamical systems where the phase space $\mathcal{U}$ is a Banach space and, in particular, a function space on a Lipschitz domain $D \subset \mathbb{R}^d$ (for finite dimensional

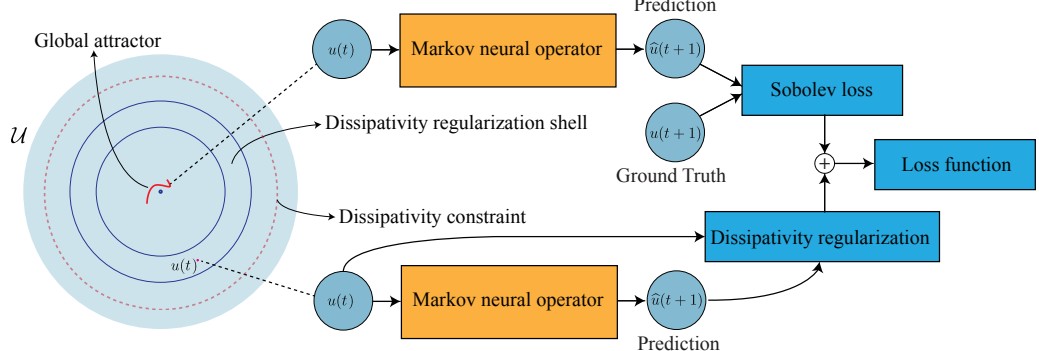

Figure 2: Markov neural operator (MNO): learn global dynamics from local data

Learning the MNO from the local time-evolution data with the Sobolev loss and dissipativity regularization. $u(t)$ is $t$'th time step of the chaotic system. Sobolev norms of various orders are used to compute the step-wise loss. Dissipativity regularization is computed by drawing a random sample $u(t)$ from the dissipativity shell to make sure that in expectation next time step prediction $\widehat{u}(t+1)$ dissipates in norm.

systems, $\mathcal{U}$ will be a Euclidean space). We are interested in the initial-value problem

$$\frac{du}{dt}(t) = F(u(t)), \qquad u(0) = u_0, \qquad t \in (0, \infty) \tag{1}$$

for initial conditions $u_0 \in \mathcal{U}$ where $F$ is usually a non-linear operator. We will assume, given some appropriate boundary conditions on $\partial D$ when applicable, the solution $u(t) \in \mathcal{U}$ exists and is unique for all times $t \in (0, \infty)$. When making the spatial dependence explicit, if it is present, we will write $u(x, t)$ to indicate the evaluation $u(t)|_x$ for any $x \in D$. We define the family of operators $S_t : \mathcal{U} \to \mathcal{U}$ as mapping $u_0 \mapsto u(t)$ for any $t \geq 0$, and note that, since (1) is autonomous, $S_t$ satisfies the Markov property i.e. $S_t(S_s(u_0)) = u(s + t)$ for any $s, t \geq 0$. We adopt the viewpoint of casting time-dependent PDEs into function space ODEs (1), as this leads to the semigroup approach to evolutionary PDEs which underlies our learning methodology.

One approach to this problem is to learn $F$ itself (e.g., [42–45]). However, in infinite dimensions, $F$ is an unbounded operator and learning unbounded operators is considerably more challenging than learning bounded operators; this is proven in [46] for linear operators. We avoid these issues by instead learning the solution operator $S_t$ for some fixed $t > 0$; this map is bounded as an operator on $\mathcal{U}$. Furthermore, large time statistical behaviour can be obtained more efficiently by using the solution operator and not resolving dynamics at timescales smaller than time $t$.

**Dissipativity.** Systems for which there exists some fixed, bounded, positively-invariant set $E$ such that for any bounded $B \subset \mathcal{U}$, there is some time $t^* = t^*(B)$ beyond which the dynamics of any trajectory starting in $B$ enters and remains in $E$ are known as **dissipative systems** [13, 14]. The set $E$ is known as the **absorbing set** of the system. For such systems, the global attractor $A$, defined subsequently, is characterized as the $\omega$-limit set of $E$. In particular, for any initial condition $u_0 \in \mathcal{U}$, the trajectory $u(t)$ approaches $A$ as $t \to \infty$. In this work, we consider dissipative dynamical systems where there exist some $\alpha \geq 0$ and $\beta > 0$ such that

$$\frac{1}{2}\frac{d}{dt}||u||^2 = \langle u, F(u) \rangle \leq \alpha - \beta||u||^2 \tag{2}$$

for all $u \in \mathcal{U}$. It can be shown that systems which satisfy this inequality are dissipative [14] with the absorbing set $E$, an open ball of radius $\sqrt{\alpha/\beta + \varepsilon}$ for any $\varepsilon > 0$. There are several well-known examples of dynamical systems that satisfy the above inequality. In this paper we consider the finite-dimensional Lorenz-63 system and the infinite-dimensional cases of the Kuramoto-Sivashinsky and 2D incompressible Navier-Stokes equations, in the form of Kolmogorov flows [13].

**Global Attractors.** A compact, invariant set $A$ is called a **global attractor** if, for any bounded set $B \subset \mathcal{U}$ and any $\epsilon > 0$ there exists a time $t^* = t^*(\epsilon, B)$ such that $S_t(B)$ is contained within an $\epsilon$-neighborhood of $A$ for all $t \geq t^*$. Formally it is the $\omega$-limit set of any absorbing set; this characterizes

it uniquely [13, 14] under assumption (2). Although unique, the $\omega$-limit set of a single initial condition may differ across initial conditions [14]. In this paper we concentrate on the ergodic setting where the same $\omega$-limit set is seen for almost all initial conditions with respect to the invariant measure. Many PDEs arising in physics such as reaction-diffusion equations (dynamics of biochemical systems) or the Kuramoto-Sivashinky and Navier-Stokes equation (fluid mechanics) are dissipative and possess a global attractor which is often finite-dimensional [13]. Therefore, numerically characterizing the attractor is an important problem in scientific computing with many potential applications.

**Data distribution.**    For many applications, an exact form for the possible initial conditions to (1) is not available; it is therefore convenient to use a stochastic model to describe the initial states. To that end, let $\mu_0$ be a probability measure on $\mathcal{U}$ and assume that all possible initial conditions to (1) come as samples from $\mu_0$ i.e. $u_0 \sim \mu_0$. Then any possible state of the dynamic (1) after some time $t > 0$ can be thought of as being distributed according to the pushforward measure $\mu_t := S_t^\sharp \mu_0$ i.e. $u(t) \sim \mu_t$. Therefore as the dynamic evolves, so does the type of likely functions that result. This further complicates the problem of long time predictions since training data may only be obtained up to finite time horizons hence the model will need the ability to predict not only on data that is out-of-sample but also out-of-distribution.

**Ergodic systems.**    To alleviate some of the previously presented challenges, we consider ergodic systems. Roughly speaking, a system is ergodic if there exists an **invariant measure** $\mu$ which is unchanged by pushforward under $S_t$ and time averages of functionals on state space $\mathcal{U}$ converge to averages against $\mu$. For dissipative systems the invariant measure is supported on the global attractor $A$ and together the pair $(A, \mu)$ capture the large time dynamics and their statistics. Proving ergodicity is hard for deterministic systems. Rigorous proofs have been developed for the Lorenz-63 model [12]; empirical evidence of ergodicity may be seen much more widely, including for the Kuramoto-Sivashinsky and Navier-Stokes equations studied in this paper. For stochastic differential equations ergodicity is easier to establish; see [47] for details.

Ergodicity mitigates learning a model that is able to predict out-of-distribution since both the input and the output of $\hat{S}_h$, an approximation to $S_h$, will approximately be distributed according to $\mu$. Furthermore, we may use $\hat{S}_h$ to learn about $\mu$ since sampling it simply corresponds to running the dynamic forward. Indeed, we need only generate data on a finite time horizon in order to learn $\hat{S}_h$, and, once learned, we may use it to sample $\mu$ indefinitely by repeatedly composing $\hat{S}_h$ with itself. Having samples of $\mu$ then allows us to compute statistics which characterize the long term behavior of the system and therefore the global attractor $A$. This strategy avoids the issue of accumulating errors in long term trajectory predictions since we are only interested in the property that $\hat{S}_h(u(t)) \sim \mu$.

## 3    Learning the Markov neural operator in chaotic dynamics

We propose the Markov neural operator (MNO), a method for learning the underlying solution operators of autonomous, dissipative, chaotic dynamical systems. In particular, we approximate the operator mapping the solution from the current to the next step $\hat{S}_h : u(t) \mapsto u(t+h)$. We approximate the solution operator $S_h$, an element of the underlying continuous time semigroup $\{S_t : t \in [0, \infty)\}$, using a neural operator as in Figure 2; in the finite dimensional case we use a feedforward neural network. See Appendix B.1 for background on the solution operator and semigroup.

**Sobolev norm as loss**    We incorporate Sobolev norms in the training process to better capture invariant statistics of the learned operator. Given the ground truth operator $S_h$ and the learned operator $\hat{S}_h$ and $f = \hat{S}_h(u) - S_h(u)$, we compute the step-wise loss in the Sobolev norm,

$$\|f\|_{k,p} = \left( \sum_{i=0}^{k} \|f^{(i)}\|_p^p \right)^{\frac{1}{p}}. \tag{3}$$

In particular, we use $p = 2$, where the Sobolev norm can be easily computed in Fourier space:

$$\|f\|_{k,2}^2 = \sum_{n=-\infty}^{\infty} (1 + n^2 + \cdots + n^{2k}) \left| \hat{f}(n) \right|^2, \tag{4}$$

where $\hat{f}$ is the Fourier series of $f$. In practice, we use $k = 0, 1, 2$ for the training.

**Long-term predictions.** Having access to the map $\hat{S}_h$, its semigroup properties allow for approximating long time trajectories of (1) by repeatedly composing $\hat{S}_h$ with its own output. Therefore, for any $n \in \mathbb{N}$, we compute $u(nh)$ as follows,

$$u(nh) \approx \hat{S}_h^n(u_0) \coloneqq \underbrace{(\hat{S}_h \circ \cdots \circ \hat{S}_h)}_{n \text{ times}}(u_0). \tag{5}$$

The above semigroup formulation can be applied with various choices of the backbone model for $\hat{S}_h$. In general, we prefer models that can be evaluated quickly and have approximation guarantees so the per-step error can be controlled. Therefore, we choose the standard feed-forward neural network [48] for ODE systems, and the Fourier neural operator [49] for infinite dimensional PDE systems.

For the the neural operator parametric class, we prove the following theorem regarding the MNO. The result states that our construction can approximate trajectories of infinite-dimensional dynamical systems arbitrary well. The proof is given in Appendix B.2.

**Theorem 1.** *Let $K \subset \mathcal{U}$ be a compact set and assume that, for some $h > 0$, the solution operator $S_h : \mathcal{U} \to \mathcal{U}$ associated to the dynamic (1) is locally Lipschitz. Then, for any $n \in \mathbb{N}$ and $\epsilon > 0$ there exists a neural operator $\hat{S}_h : \mathcal{U} \to \mathcal{U}$ such that*

$$\sup_{u_0 \in K} \sup_{k \in \{1, \ldots, n\}} \|u(kh) - \hat{S}_h^k(u_0)\|_{\mathcal{U}} < \epsilon.$$

Theorem 1 indicates that if the backbone model is rich enough, it can approximate many chaotic systems for arbitrarily long periods. For finite-dimensional systems, the same theorem holds with feed-forward neural networks instead of neural operators. We note that standard neural networks such as RNNs and CNNs *do not possess* such approximation theorems in the infinite-dimensional setting.

**Invariant statistics.** A useful application of the solution operators is to estimate statistics of the invariant measure of a chaotic system. Assume the target system is ergodic and there exists an invariant measure $\mu$ as discussed in Section 2. An invariant statistic is defined as

$$T_G \coloneqq \int_{\mathcal{U}} G(u) \, \mathrm{d}\mu(u) = \lim_{T \to \infty} \frac{1}{T} \int_0^T G(u(t)) \, \mathrm{d}t \tag{6}$$

for any functional $G : \mathcal{U} \to \mathbb{R}^d$. Examples include the $L^2$ norm, any spectral coefficients, and the spatial correlation, as well as problem-specific statistics such as the turbulence kinetic energy and dissipation rates in fluid flow problems. Given the property (5) and using the ergodicity from (6), the approximate model $\hat{S}_h$ can be used to estimate any invariant statistic simply by computing $T_G \approx \frac{1}{n} \sum_{k=1}^n G(S_h^k(u_0))$ where $nh = T$ and $T > 0$ large enough.

**Encouraging dissipativity via regularization.** In practice, training data for dissipative systems is typically drawn from trajectories close to the global attractor, so a priori there is no guarantee of a learned model's behavior far from the attractor. Thus, if we seek to learn the global attractor and invariant statistics of a dynamical system, it is crucial that we encourage learning dissipativity. To learn $\hat{S}_h : u(t) \mapsto u(t + h)$ we propose the following loss function, found by supplementing the Sobolev loss (3) with a dissipativity-inducing regularization term:

$$\underbrace{\int_{\mathcal{U}} \|\hat{S}_h(u) - S_h(u)\|_{k,2}^2 \, \mathrm{d}\mu(u)}_{\text{step-wise Sobolev loss}} + \alpha \underbrace{\int_{\mathcal{U}} \|\hat{S}_h(u) - \lambda u\|_{\mathcal{U}}^2 \, \mathrm{d}\nu(u)}_{\text{dissipativity regularization}}. \tag{7}$$

Here $\mu$ is the underlying distribution of the training data, $\alpha$ is a loss weighting hyperparameter, and $0 < \lambda < 1$ is some constant factor for scaling down (i.e., enforcing dissipativity) inputs $u$ drawn from a probability measure $\nu$. We choose $\nu$ to be a uniform probability distribution supported on some shell with a fixed inner and outer radii from the origin in $\mathcal{U}$; these are chosen so that the support of $\nu$ is disjoint from the attractor. Our dissipative regularization term scales down $u$ by constant $\lambda$, but in principle alternative dissipative cost functionals can be used.

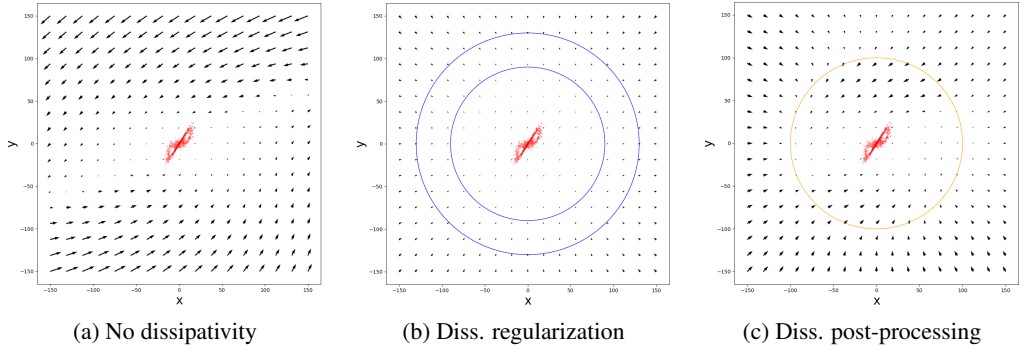

|(a) No dissipativity | (b) Diss. regularization | (c) Diss. post-processing|

Figure 3: Dissipativity regularization on the Lorenz 63 system – flow maps.
Red points are training data on the attractor. The dissipativity regularization in (b) is imposed by augmenting the data on the blue shell. (c) Post-processing with $\alpha = 100$ (the orange circle) and $\beta = 0.1$ as in eq. (9).

**Enforcing dissipativity via post-processing.**    While the previous dissipativity regularization enforces a dissipative map in practice, it does not readily yield to theoretical guarantees of dissipativity. For an additional safeguard against model instability, we enforce a hard *dissipativity constraint* far from the attractor and from the shell where $\nu$ is supported, resulting in provably dissipative dynamics.

In detail, we post-process the model: whenever the dynamic moves out of an a priori defined stable region, we switch to the second model $\Psi$ that pushes the dynamic back. The new model combines the learned model $\hat{S}_h$ and the safety model $\Psi$, via a threshold function $\rho$ :

$$\hat{S}_h'(u) = \rho(\|u\|)\hat{S}_h + (1 - \rho(\|u\|))\Psi(u), \tag{8}$$

where $\Psi$ is some dissipative map and $\rho$ is a partition of unity. For simplicity we define

$$\Psi(u) = \lambda u \qquad \rho(\|u\|) = \frac{1}{1 + e^{\beta(\|u\|-\alpha)}}, \tag{9}$$

where $\alpha$ is the effective transition radius between $\hat{S}_h$ and $\Psi$ and $\beta$ controls the transition rate. Note that this choice of $\Psi$ is consistent with the regularization term in the loss (7).

By its construction, the post-processed model is dissipative. However, its dynamics (in particular the unlearnable transition between $\hat{S}_h$ and $\Psi$) are not as smooth as the dissipative regularization model, which learn the whole dynamic together, as shown in Figure 3. Combining regularization and post-processing results in a dissipative model with both smooth dynamics and theoretical guarantees.

## 4    Experiments

We evaluate our approach on the finite-dimensional, chaotic Lorenz-63 system as well as the chaotic 1D Kuramoto-Sivashinsky and 2D Navier-Stokes equations. In all cases we show that encouraging dissipativity is crucial for capturing the global attractor and evaluating statistics of the invariant measure. To the best of our knowledge, we showcase the first machine learning method able to predict the statistical behavior of the incompressible Navier-Stokes equation in a strongly chaotic regime.

### 4.1    Lorenz-63 system

To motivate and justify our framework for learning chaotic systems in the infinite-dimensional setting, we first apply our framework on the simple yet still highly chaotic Lorenz-63 ODE system, a widely-studied [50] simplified model for atmospheric dynamics given by

$$\dot{u}_x = \alpha(u_y - u_x), \qquad \dot{u}_y = -\alpha u_x - u_y - u_x u_z, \qquad \dot{u}_z = u_x u_y - b u_z - b(r + \alpha). \tag{10}$$

We use the canonical parameters $(\alpha, b, r) = (10, 8/3, 28)$ [51]. Since the solution operator of the Lorenz-63 system is finite-dimensional, we learn it by training a feedforward neural network on a single trajectory with $h = 0.05s$ on the Lorenz attractor. Figure 3 shows that dissipativity regularization produces predictions that isotropically point towards the attractor. Observe that the our network is also dissipative outside the regularization shell.

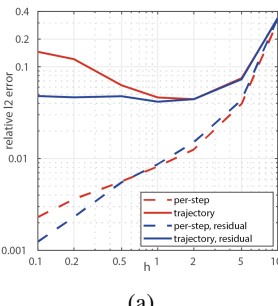 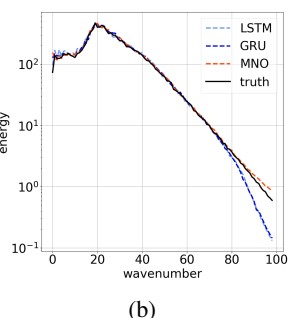 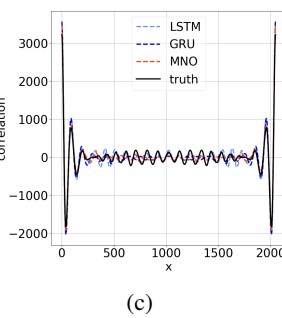

(a)                   (b)                   (c)

Figure 4: Choice of time step (a), Fourier spectrum (b), and spatial correlation (c) for KS equations. (a) Both learning solution operator and learning identity residual induce smaller per step error for smaller $h$. When learned models are composed to generate longer trajectories, we observe that learning residual is advantageous. (b) Fourier spectrum of the predicted attractor. All models capture Fourier modes with magnitude larger than $O(1)$, while MNO is more accurate on the tail. (c) Spatial correlation of the attractor, averaged in time. MNO is more accurate on near-range correlation, but all models miss long-range correlation.

We empirically find that dissipativity can be encouraged without significantly affecting the model's step-wise relative $L^2$ error and the learned statistical properties of the attractor. We also find that varying the dissipativity hyperparameters (loss weight $\alpha$, scaling factor $\lambda$, and shell radius) does not significantly affect the step-wise or dissipative error. Details in Appendix A.1.

## 4.2   Kuramoto-Sivashinsky equation

We consider the following one-dimensional KS equation,

$$
\frac{\partial u}{\partial t} = -u\frac{\partial u}{\partial x} - \frac{\partial^2 u}{\partial x^2} - \frac{\partial^4 u}{\partial x^4}, \qquad\qquad \text{on } [0, L] \times (0, \infty),
$$
$$
u(\cdot, 0) = u_0, \qquad\qquad \text{on } [0, L],
$$

where the spatial domain $[0, L]$ is equipped with periodic boundary conditions. We study the impact the time step $h$ has on learning. Our study shows that when the time steps are too large, the correlation is chaotic and hard to capture. But counter-intuitively, when the time steps are too small, the evolution is also hard to capture. In this case, the model's input and output are very close, and the identity map will be a local minimum. We thus use the MNO to also learn the time-derivative or residual. Figure 4a shows the results for varying $h$ and when MNO is used to learn either the identity residual or the solution operator. We observe that the residual model has a better per-step error and accumulated error at smaller $h$. When the time step is large, there is no difference in modeling the residual.

As shown in Figures 4b and 4c, we compare the performance of the MNO model against LSTM and GRU that we use to model the evolution operator of the KS equation with $h = 1s$. We observe that MNO model accurately recovers the Fourier spectrum of KS equation. For all other statistics (see Appendix A.2) all models perform similarly, except on the velocity distribution where MNO performs best. We emphasize that some of these statistics are very challenging to capture and most machine learning approaches in the literature thus far fail to do so (see Appendix A.2 for details).

## 4.3   Kolmogorov Flow

We consider two-dimensional Kolmogorov flow (a form of the Navier-Stokes equations) for a viscous, incompressible fluid,

$$
\frac{\partial u}{\partial t} = -u \cdot \nabla u - \nabla p + \frac{1}{Re}\Delta u + \sin(ny)\hat{x}, \qquad \nabla \cdot u = 0, \qquad \text{on } [0, 2\pi]^2 \times (0, \infty) \qquad (11)
$$

with initial condition $u(\cdot, 0) = u_0$ where $u$ denotes the velocity, $p$ the pressure, and $Re > 0$ is the Reynolds number. We encourage dissipativity during training with the criterion described in eq. (7), with $\lambda = 0.5$ and $\nu$ being a uniform probability distribution supported on a shell around the origin.

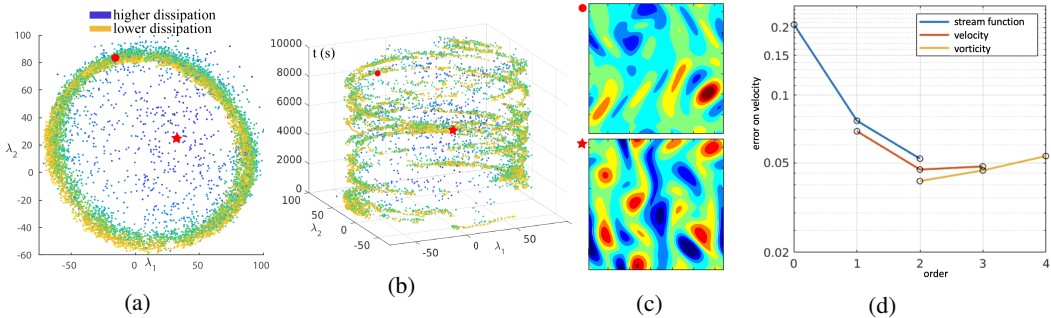

(a)               (b)                    (c)               (d)

Figure 5: The learned attractor of Kolmogorov flow ($Re = 500$) and choice of Sobolev loss for KF. (a-c) The 10000 time steps trajectory generated by MNO projected onto the first two principal components. Each point corresponds to a snapshot on the attractor. The vorticity field for two points is shown. (d) Velocity error for models trained on stream function, velocity, and vorticity using Sobolev loss of different orders.

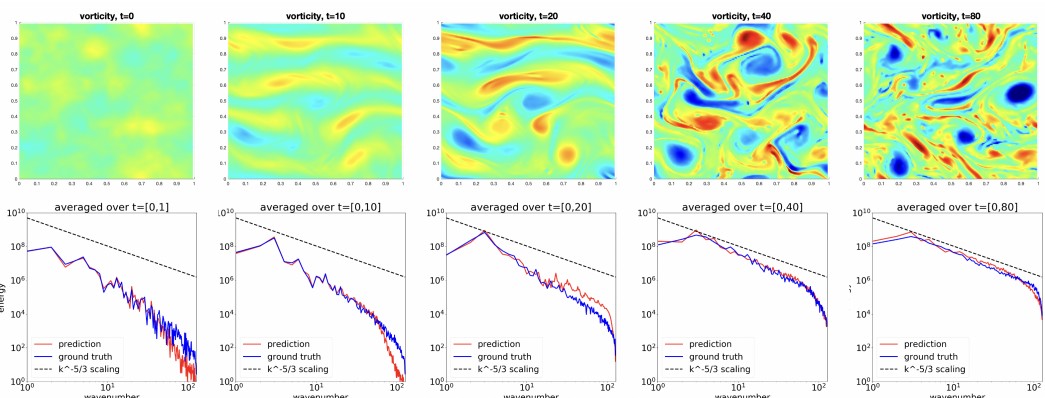

Figure 6: Simulation of Kolmogorov flow ($Re = 5000$) with the dissipative MNO model. The first column corresponds to the initial condition and is sampled from a Gaussian random field. Around $t = 10$ to $t = 20$, we see the energy injected from the source term $\sin(4y)$ (12), and also the energy transfers from higher frequencies to the lower frequencies. The dotted line is the Kolmogorov energy cascade rate.

We test the effect of encouraging dissipativity in the turbulent (and blow-up prone) $Re = 500$ setting, where we observe that a non-dissipative model blows up when composed with itself multiple times if the initial condition is perturbed slightly from the attractor (Figure 1), even though the model achieves relatively low $L^2$ error. In contrast, we empirically observe that the dissipative MNO does not blow up and its composed predictions returns to the attractor even when the initial condition is perturbed.

**Simulation on turbulent $Re = 5000$ case.** The proposed dissipative MNO model can stably learn complex chaotic dynamics at high Reynolds numbers. As shown in Figure 6, the model captures the energy spectrum that converges to the Kolmogorov energy cascade rate of $k^{-5/3}$. The learned model is dissipative with the dissipative regularization, without compromising the accuracy as shown in Table 5. Even given a $O(10^4)$-scaled perturbation, the dynamic quickly returns to the attractor of the system. In contrast, the model without dissipative regularization will stay outside the attractor given even a $O(10)$-scale perturbation.

**Accuracy with respect to various norms.** We study performance of MNO along with other predictive architectures, U-Net [39] and LSTM-CNN [40], on modeling the vorticity $w$ in the relatively non-turbulent $Re = 40$ case 3. The MNO achieves one order of magnitude better accuracy compared to this architectures. As shown in Table 3, we train each model using the balanced $L^2(= H^0)$, $H^1$, and $H^2$ losses, defined as the sum of the relative $L^2$ loss grouped by each order of derivative. We measure the error with respect to the standard (unbalanced) norms. The MNO with $H^2$ loss consistently achieves the smallest error on vorticity for all of the $L^2$, $H^1$, and $H^2$ norms. However, the $L^2$ loss achieves the smallest error on the turbulence kinetic energy (TKE), while the

$H^1$ loss achieves the smallest error on the dissipation $\epsilon$. Vorticity fields for NS equation ($Re = 40$) are shown in Figure 11. The figure indicates that the predicted trajectories (b) (c) (d) share the same behaviors as in the ground truth (a), indicating that the MNO model is stable.

**Estimating the attractor and invariant statistics.** We compose MNO 10000 times to obtain the global attractor, and we compute the PCA (POD) basis of these 10000 snapshots and project them onto the first two components. As shown in Figure 5a, we obtain a cycle-shaped attractor. The true attractor has dimension on the order of $O(100)$ [13]. If the attractor is a high-dimensional sphere, then most of its mass concentrates around the equator. Therefore, when projected to low-dimension, the attractor will have a ring shape. We see that most points are located on the ring, while few points are located in the center. The points in the center have high dissipation, implying they are intermittent states. In Figure 5b we add the time axis. While the trajectory jumps around the cycle, we observe there is a rough period of 2000s. We perform the same PCA on the training data, showing the same behavior. Furthermore, in Figure 12, we present invariant statistics for the NS equation ($Re = 40$), computed from a large number of samples from our approximation of the invariant measure. We find that the MNO consistently outperforms all other models in accurately capturing these statistics.

**Derivative orders.** Roughly speaking, vorticity is the derivative of velocity; velocity is the derivative of the stream function. Therefore we can denote the order of derivative of vorticity, velocity, and stream function as 2, 1, and 0 respectively. Combining vorticity, velocity, and stream function, with $L^2$, $H^1$, and $H^2$ loss, we have in total the order of derivatives ranging from 0 to 4. We observe, in general, it is best practice to keep the order of derivatives in the model at a number slightly higher than that of the target quantity. For example, as shown in Figure 5d, when querying the velocity (first-order quantity), it is best to use second-order (modeling velocity plus $H^1$ loss or modeling vorticity plus $L^2$ loss). This is further illustrated in Table 4 for the $Re = 500$ case. In general, using a higher order of derivatives as the loss will increase the power of the model and capture the invariant statistics more accurately. However, a higher-order of derivative means higher irregularity. It in turn requires a higher resolution for the model to resolve and for computing the discrete Fourier transform. This trade-off again suggests it is best to pick a Sobolev norm not too low or too high.

## 5 Conclusion

In this work, we propose a machine learning framework that trains from local data and enforces dissipative dynamics. Experiments also show MNO predicts the attractor which shares the same distribution and invariant statistics as the true function space trajectories. The simulations achieved by the MNO have the potential to further our understanding of many physical phenomena and the mathematical models that underlie them.

Furthermore, the MNO shows great potential for modeling partially observed systems or studying systems that exhibit bifurcations, both of which are of great interest for engineering applications. The study of non-ergodic dissipative systems and their basins of attraction is also an important direction for future study.

## Acknowledgements

Z. Li gratefully acknowledges the financial support from the Kortschak Scholars, PIMCO Fellows, and Amazon AI4Science Fellows programs. M. Liu-Schiaffini is supported by the Stephen Adelman Memorial Endowment. A. Anandkumar is supported in part by Bren endowed chair. K. Bhattacharya, N. B. Kovachki, B. Liu, A. M. Stuart gratefully acknowledge the financial support of the Army Research Laboratory through the Cooperative Agreement Number W911NF-12-0022. A. M. Stuart is also grateful to the US Department of Defense for support as a Vannevar Bush Faculty Fellow. Research was sponsored by the Army Research Laboratory and was accomplished under Cooperative Agreement Number W911NF-12-2-0022. A part of this work took place when K. Azizzadenesheli was at Purdue University. The views and conclusions contained in this document are those of the authors and should not be interpreted as representing the official policies, either expressed or implied, of the Army Research Laboratory or the U.S. Government. The U.S. Government is authorized to reproduce and distribute reprints for Government purposes notwithstanding any copyright notation herein.

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
