# OpenReview forum: "Learning Chaotic Dynamics in Dissipative Systems"
_NeurIPS.cc/2022/Conference — NeurIPS 2022 Accept_

### Official Review · Reviewer_Hc1s · 2022-06-27

**Rating:** 5
**Confidence:** 5
**Soundness:** 3 good
**Presentation:** 3 good
**Contribution:** 4 excellent

**Summary:**

This is a very interesting paper and addresses an important challenge in terms of long-term stability of high-dimensional turbulent flow. The authors show that adding a dissipative term in the loss and training it with sobolev loss would improve the stability of the autoregressive data-driven models (here they show that a Markov neural operator) is the best skeleton model.


**Questions:**

The paper discusses and investigates realistic metrics pertinent to the problem and I've thoroughly enjoyed reading it. However, there are still some questions/clarifications that remain for the authors to justify their claims. Herein, I summarize some of the main issues that I find. Beyond that, there is a large class of literature in weather/climate modeling and fluid dynamics that talks about these issues at large; I suggest that the authors look at some of the data-driven weather prediction papers that talk about long-term instability. Adding that context to this paper would make it even more meaningful for readers who would actually care about why this is an important problem for the physical sciences.


1. Line 2. -- The issue is not just with time-stepping, but with the fact that because of a positive Lyapunov exponent the initial perturbation grows exponentially.

2. Line 19-20. There seems to be other work (one at least) that looks at long-term statistics of chaotic systems, e.g. PDFs including the tails that capture extreme events. Even more so, it turns out RCs can do both short- and long-term seamlessly. https://npg.copernicus.org/articles/27/373/2020/. Surely, there could be other papers that do the same. The authors' statement is not accurate here. They are however correct when they say that RCs definitely do not scale for more complex problems. Even so, the authors are still looking at toy systems that RCs excel at (except for the NS case).

3. Line 25-26 What's the rationale behind calling it "ill-posed", citations to relevant papers are essential.

4. Line 40. Typo. charges->changes

5. For Lorenz 63, simpler methods are long-term stable, even for KS, and for other more complex Lorenz-type systems as shown above. The first two examples are not interesting or relevant when we want to talk about long-term statistics. 2D Kolomogrov is a great example, and there are many other relevant problems that have been studied for long-term stability along those lines, e.g., QG turbulence, other climate models, and real weather data. There is a big community working on this specific problem. The authors need to look at that literature:
(i) https://arxiv.org/abs/2205.04601
(ii) https://arxiv.org/abs/2202.11214
(ii) https://arxiv.org/abs/2202.07575
(iv) https://dl.acm.org/doi/10.1145/3429309.3429325
(v) https://agupubs.onlinelibrary.wiley.com/doi/full/10.1029/2020MS002109


In the purely ML community, there are a few other papers as well (who tried and improved stability but eventually couldn't make it infinitely stable) for Kolomogrov turbulence (more complicated cases than the current paper). The manuscript would benefit from putting the problem in the context of this bigger goal.
 https://arxiv.org/abs/2112.15275

6. The effect of "h" is very interesting in the KS problem. There are a few other papers that have found results that might be contradicting this specific paper, or maybe not if I have misunderstood. But either way, that's good. It generates discussion and the importance of exploring this problem more carefully. However, I feel that the authors should acknowledge those papers as well. While not done in the context of operator modeling, they have been done on more complex, realistic, and practical problems of chaos and turbulence. Here are a few of those papers :

(i) https://royalsocietypublishing.org/doi/10.1098/rsta.2021.0200 (Section 2)
(ii) https://agupubs.onlinelibrary.wiley.com/doi/epdf/10.1029/2020MS002203 (Figure 2)
(ii) https://gmd.copernicus.org/articles/15/2221/2022/ (Section 4.3)

7. I appreciate how the authors interpret stability as something beyond just running the model and not getting "NaN". Stability means that the flow is both physical and non-drifting and has the right variability. However, I am still not 100% convinced that this model can serve as a real long-term stable model for a real complex flow. Here are a few things the authors can show to remove my doubts:
    --- While the model captures the spectrum quite well, it deviates in the high-wavenumbers, an essential reason for instability (Figure 4b). Can the authors show how the spectrum deviates as we move forward in time? Basically, take the best model and plot the output's spectrum at various time steps during prediction and compare it with the truth. Furthermore, I don't think that KS is that complicated a system, wherein "spectral bias" (the fact that you lose high wavenumbers as you integrate in time) is that big of a problem. However, in the N-S case with high Re, this would become a definite problem especially because you have inverse cascade. Also, for KS, I think Pathak et al, PRL, had long-term statistics, without doing anything? Does this KS have the same dimension or is it higher?
   ----  It's great that the authors look at the PC1 autocorrelation curve. It must be pointed out that the authors' model does not capture it, or am I mistaken? I also understand that it's difficult to capture PC1 autocorrelation in KS because, in reality, I don't think PC1 in this system has any physical meaning (e.g., in a real system like Earth's climate PC1 has connections to ENSO and so on..). Still the fact that the model doesn't go absolutely haywire is promising. Yet, the authors should clarify this
  ---  I don't think the PDF is also captured correctly. Please plot PDF in semilog, so that the tails (extreme events) are more prominent, that's what matters. But here, I think even the bulk is wrong in some portions. It's still impressive that you have the right range of values.

8.  For the N-S case, I have some major confusion. In the main paper (not the appendix), the spectrum is not plotted. In the appendix, I see some images with Re=40. Re=40 is not nearly turbulent enough. That's hardly interesting in the context of turbulence and it's long-term predictability. Even Re=500 is not a realistic set-up. I must say here, that I understand the difficulty to scale up the turbulent flow to high Re, it's both difficult computationally, and probably very hard to predict. But based on the context of the problem, if this model really is going to serve practical problems in terms of long-term stability, the 2D N-S case should have Re=10000 or higher. That's the regime when we have all the complexities of this flow including when the effect of inverse cascade would be prominent. Here're a few papers that look at NS cases in realistic regimes.
    (i) https://www.pnas.org/doi/10.1073/pnas.2101784118
    (ii) https://www.sciencedirect.com/science/article/pii/S0021999122001528
    (iii) https://www.cambridge.org/core/journals/journal-of-fluid-mechanics/article/abs/subgrid-modelling-for-twodimensional-turbulence-using-neural-networks/10EDED1AEAA52C35F3E3A3BB6DC218C1

9. For the N-S case, the energy spectrum at different time steps needs to be shown for the best model and the baseline. That would clearly show how well the spectrum is captured. It would also only be interesting for a case where the flow is turbulent (Re>10000)

10. Again, the PDFs should be shown in semilog plots to highlight the tails. Also, what's going on with the PDF of KE for the N-S case? It's not really getting captured right? In this case, as well, there's no dominant mode of variability, so PC1 does not have any meaning right? But still, plot the true PC1 autocorrelation plot to show how well/poorly it's captured.

11. The authors might benefit from looking at another system. A system that has a dominant mode and coherent structures while being turbulent enough. I think the discussion around stability would have far more importance there with PC1, autocorr, etc even if they are not perfectly captured.

Finally, despite my long review and many questions, I enjoyed reading the paper. The authors have done a great job in addressing the question of stability "correctly"  by looking at the right set of metrics. While there are these lingering questions, once properly addressed this would be an important contribution to the ML and physics community.

**Strengths And Weaknesses:**

The major strength of this paper is that it addresses a difficult problem in the physical sciences that have a lot of stakeholders and look at pertinent and physically justified metrics to support their claim.

However, the major weakness of this work is that 2 of the 3 systems considered are not realistic enough to show issues with instability (the problem at hand) and the 3rd system is also not the best realistic case. Other papers have looked at more complex systems. The weakness still does not take away a very good set of analysis and transparent results that the authors have shown. Still it needs some work before getting accepted.

---

> ### Author Response · Authors · 2022-08-02
> **Response to Reviewer Hc1s**
>
>
> We appreciate your insightful comments and we are glad that the reviewer enjoyed reading our paper. The two main changes we have made to address your points are (1) including MNO results on Kolmogorov Flow ($Re=5000$) and (2) including a method for enforcing dissipativity via post-processing, which results in dynamics that are *provably* stable, in contrast to previous works.
>
> We now address each of your points in detail:
>
> 1. We have changed the wording here; see lines 16-17.
> 2. Thanks for pointing out the reference. We have changed the wordings here.
> 3. We agree: "ill-posed" may not have been the best phrase here; we have changed this sentence in the revised version.
> 4. Thanks, we have changed the wording in the introduction.
> 5. Thanks for the references; we have included them in our paper. Although these papers explicitly address the fact that long-term stability is a desirable feature, they do not provide a principled data-driven modeling paradigm to enforce it.
>
>      In this paper we work within a mathematical setting that incorporates a variety of problems, including our three examples, and demonstrate two principled methods to enforce dissipativity: (1) by augmenting the training procedure; (2) by post-processing standard methodologies. Furthermore the method in (2) leads to provably dissipative (in our precisely defined sense) models, something that has not been achieved in the literature before.
> 6. Thanks for the references on varying the choice of $h$; we have cited these in our paper.
> 7. Thanks for your detailed comments. Our goal in this work is not to address large complex models directly, but rather to introduce---on a few test problems---new methodologies to ensure dissipativity which can then be deployed in more complex models.
>
>     To address your concerns, we have plotted the vorticity spectrum over time (also in a higher Reynolds number) in Figure 6. The model captures the energy spectrum that converges to the Kolmogorov energy cascade rate of  $k^{-5/3}$.
>
>     As the reviewer pointed out, none of the models capture the auto-correlation of the PCA modes, probably because the PCA modes does not really have any physical meaning in this case. We will make sure to plot the probablity density functions in the semilog scale in the camera ready version.
>
>
> 8. Thank you for the references. Kochkov et. al. study DNS at $Re=1000$ and $4000$, and LES at $Re=10^5$; Guan et. al. study LES; Maulik et. al. study the local closure at $Re=10^5$. To match their setting, we add a DNS at $Re=5000$ to show our model's performance under more complex conditions, and we have plotted the vorticity spectrum over time. Since we are learning the solution operator, it makes it easier to scale to more complicated systems with higher Reynolds numbers. The main constraint is to generate the datasets with DNS.
>
>     ![](https://i.imgur.com/DTutGuU.jpg)
>
>     We want to emphasize our goal in this work is not to show that we can simulate turbulence, but rather to introduce a new framework to systematically learn the attractor of chaotic systems and ensure dissipative learned dynamics. We test our methods on test problems (e.g., Lorenz-63 being a simple baseline to visualize the results).
>
> 9. We have added the evolution of the energy spectrum at Figure 6.
> 10. As the reviewer pointed out, none of the models capture the distribution of TKE exactly, but MNO is closer. The ground truth kinetic energy has two peaks. Both the MNO and UNet models predict one peak. Still the MNO model is more accurate compared to UNet. Similar to the KS equaiton, the PCA modes of NS are hard to capture too. It's probably because PCA modes do not really have any physical meaning neither. We will make sure to plot the probablity density functions in the semilog scale in the camera ready version.
> 11. Thanks for the suggestion. We are happy to try another turbulent system that has a dominant mode and coherent structures. Suggestions are welcome. Meanwhile we would like to re-emphasize that our goal in this work is to introduce new methodologies for learning chaotic systems (enforcing dissipativity, training with Sobolev norms, etc.), which can later be applied to more complex systems. However, we hope you find interesting our MNO results on Kolmogorov Flow with $Re=5000$, as this is a more complex flow than for $Re=500$.

---

> > ### Comment · Reviewer_Hc1s · 2022-08-10
> > **Comments on authors' response**
> >
> > 7. I appreciate the authors' motivation to address this problem from a principled approach. My point was that for the first two systems, the instability problem does not exist with simpler RC-like methods can get stable solutions. Hence, they are not suitable test cases. For the third system, the authors' analysis seems fine. I'm a bit confused about the slope of the spectrum: it should be k^-3 not k^(-5/3). The slope of -5/3 is for 3D turbulence when there is a direct cascade. There's something not right here.
> >
> > 8. In the new figure that the authors added it seems that the energy at the smallest scales is increasing with time. This is a stationary problem, if the model is really stable the spectrum should remain the same (at least the structure), but a significant curl-up can be seen in the figure.
> >
> > I still think that this is an interesting contribution. Obviously, this is a challenging problem, but the approach is rigorous and in my opinion, in the correct direction as well. For that, I would increase my score and recommend that this paper gets accepted.
> >
> > Please fix the slope of the spectrum --- something is definitely wrong there ( take a look at this: https://courses.physics.ucsd.edu/2019/Winter/physics116_216/annurev-fluid-120710-101240.pdf) and I recommend that the authors plot the true DNS spectrum on top of the predicted spectrum in Fig 6 (or whatever is going to be the figure number for the Re=5K figure)  showing the curl up. I also recommend removing the Re=400 figure from the main paper and replacing it with the Re=5K results. There is no value in looking at such a simple system and talk about instability. Even if the Re=5K results look relatively poor, it is a truly challenging case and a physics audience -- the kind that understands turbulence--would appreciate it.

---

> > > ### Author Response · Authors · 2022-08-10
> > > **Response to Reviewer Hc1s**
> > >
> > > Thank you for your response. Your feedback helped us make the paper stronger!
> > >
> > > The authors-reviewers discussion period may have ended, but allow us to post a brief response regarding the slope.
> > >
> > > 7. If we understand correctly, the slope of spectrum is $k^{-5/3}$ in the inverse cascade range ($k_a << k << k_f$), and $k^{-3}$ in the direct-cascade range $(k>>k_f)$, which is exactly written in your reference (page 431, equation 12 & 13 https://courses.physics.ucsd.edu/2019/Winter/physics116_216/annurev-fluid-120710-101240.pdf). In our setting, we use 128x128 grid in the model, so we only compute the wavenumber k up to 100. It is possible $k_f > 100$, so the $k^{-3}$ range does not show up. We will certainly add the spectrum of the ground truth DNS simulation in the paper.
> > >
> > > 8. We initialize the system from a Gaussian random field that has low initial energy. The energy is injected from the force term over time, so we can observe energy at the small scales is increasing and converging to a stable state (turbulence builds up).

---

### Official Review · Reviewer_MCtx · 2022-07-06

**Rating:** 5
**Confidence:** 3
**Soundness:** 3 good
**Presentation:** 3 good
**Contribution:** 3 good

**Summary:**

This paper proposes a machine learning framework (MNO) by using neural networks or Fourier neural operators to learn the underlying Markov operator of dissipative chaotic systems. Numerical results show that MNO has lower loss and outperforms the other neural network models, including U-Net, LSTM-CNN, and GRU.

**Questions:**

1. Could you provide the formula of Sobolev Loss and Dissipativity Loss in Figure 2. I guess that the Dissipativity Loss is given by Equation 5. Then, is the ground truth exactly match the Equation 5? If not, does "lower loss" make sense? I think it might be more appropriate to treat it as a regular term.

2. How about using continuous models to learn continuous dynamics？For example, Neural ODE or its structure-preserving extensions [1]. In [1], they also consider learning Lorenz-63 system. For infinite dimensional PDE systems, in my view, we can directly replace the neural network in a continuous model by a neural operator to obtain a infinite continuous model.

[1] Haijun. Yu, Xinyuan. Tian, Weinan. E, Qianxiao. Li, Onsagernet: Learning stable and interpretable dynamics using a generalized onsager principle, PHYSICAL REVIEW FLUIDS 6, 114402 (2021)


**Limitations:**

Yes, the authors addressed the limitations.

**Strengths And Weaknesses:**

This paper proposes to learn the underlying Markov operator of dissipative chaotic systems using deep learning. Several architectures are investigated in detail. The results is interesting, however, I'm not sure whether learning Markov operator of dissipative chaotic systems is relevant for deep learning. In my view, this paper is more suitable for some dynamical journals, such as SIAM Journal on Applied Dynamical Systems, Chaos, ... .

---

> ### Author Response · Authors · 2022-08-02
> **Response to Reviewer MCtx**
>
>
> Thanks for your comments. Due to the recent advances in deep learning, the problem of learning chaotic dynamics  from data (e.g., modeling turbulent flows for weather applications or for control) has become of major importance in machine learning and in applied science in general.
>
> In our work, we propose a systematic framework for learning chaotic dynamics in realistic dissipative systems. We believe that our work sits at the intersection of research in dynamical systems and the modern advances of deep learning, and we advocate for continued communication between the fields. Reviewers 3spS, NdmN, and Hc1s all show interest in this work, which suggests Neurips can be a proper venue.
>
> 1. We appreciate the suggestions. We have defined our Sobolev norm step-wise loss term (eqs. 5-6) and the overall loss function used to train our MNO (eq. 5). We have also renamed the "dissipativity loss" to "dissipativity regularization." See Figure 2.
>
> 2. Learning continuous dynamics is certainly an interesting avenue of study, with much prior work focused in that direction (e.g., SINDy, Neural ODE, OnsagerNet, etc.). This amounts to learning $F$ in eq. 1 of our paper.
>
>     However, there are two reasons why we wish to instead learn the solution operator for the system: (1) If we learn $F$, we still need an integrator to compute predictions over long time intervals, which is computationally expensive, (2) In infinite dimensions, $F$ is unbounded, and learning unbounded operators is a considerably more challenging problem. [de Hoop et al., 2021] show, in the context of linear operators, that learning is easier (in terms of sample complexity) for compact and bounded operators than for unbounded operators.
>
>     To clear up confusion, we have added this point more explicitly in our paper. See Section 2, paragraph 2.

---

> > ### Comment · Reviewer_MCtx · 2022-08-07
> > **Response to Authors**
> >
> > I have read the rebuttal and I would like to thank the authors for their response.
> >
> > I have increased my score accordingly. I see another review explicitly asks for learning continuous dynamics. Whereas the response gave an explanation that “learning continuous dynamics needs an integrator to compute predictions over long time intervals, which is computationally expensive," this point needs to be quantified, as we generally use small-scale neural networks, and only a few steps are required as a substitute for one-step prediction of a discrete model when data step size is small.

---

> > > ### Author Response · Authors · 2022-08-09
> > > **Response to Reviewer MCtx**
> > >
> > > We really appreciate your positive feedback. These comments encourage us and help us to improve our paper. We agree that there may be a trade-off between the numbers of parameters required to represent the solution operator and the time over which the solution operator is represented; quantifying this trade-off, and including the continuous-time limit and vector field driving it, will be of value and almost certainly needs to be addressed on a case by case basis. However, for PDEs, where the operator learning problem is for an unbounded operator, our numerical results strongly suggest the value of learning the solution operator map at a positive time, and not its infinitesimal generator.
> > >
> > > Learning continuous dynamics for complex PDE systems is not only costly, but also limited for two reasons: (1) the time-derivative (infinitesimal generator) is an unbounded operator, which is more difficult to learn. (2) the continuous dynamics only applies to finite-dimensional ODE system. For PDE systems, the integrator (such as the forward-time centered-space method, etc.) needs to incorporate the spatial derivatives, which are usually unavailable in the reduced-order models. These works such as SINDy have to discretize the space using POD/PCA, which limits its performance on complex systems with high-dimensional attractors such as the Navier-Stokes equation.
> > >
> > > As a comparison, the foundational work [Discovering governing equations from data by sparse identification of nonlinear dynamical systems](https://www.pnas.org/doi/10.1073/pnas.1517384113) considered a laminar cylinder flow with Re=100. It simplifies the complicated system into only 3 modes: two PCA modes plus the shift mode. The work is the first to generate attractors [Fig 2] similar to our [Figure 5. ab], but this simplified model cannot simulate the full-field velocity and spectrum as we did [Figure 6,9,10,11].
> > >
> > > Reviewer NdmN points out the works by the Brunton group or Edward Ott’s group to argue that "The authors are not the first to put the focus on invariant statistics." Specifically,  of the three papers below learning continuous dynamics, none of them consider complex chaotic PDE systems such as the Navier-Stokes equation (with Reynolds number up to 5000, like we do):
> > > 1. [Discovering Governing Equations from Partial Measurements with Deep Delay Autoencoders](https://arxiv.org/pdf/2201.05136.pdf): This paper combines deep learning to uncover effective coordinates and the sparse identification of nonlinear dynamics (SINDy) for interpretable modeling. They focus on ODE system such as **Lorenz**.
> > > 2. [Data-driven discovery of coordinates and governing equations](https://arxiv.org/pdf/1904.02107.pdf): Similarly, the paper combines the strengths of deep neural networks for flexible representation and sparse identification of nonlinear dynamics (SINDy) for parsimonious models. They focus on simpler systems such as **Lorenz**, **Reaction-diffusion**, and **Nonlinear pendulum**.
> > > 3. [SINDy-BVP: Sparse Identification of Nonlinear Dynamics for Boundary Value Problems](https://arxiv.org/pdf/2005.10756.pdf): This work main considers boundary value problems (BVPs) such as **Sturm-Liouville**, **Poisson equation** and **Euler-Bernoulli Beam theory**.
> > >
> > > Similarly, there are few works on reservoir computing or Koopman operators can handle such non-linear systems. If you have any specific model in mind that can run on the turbulent Navier-Stokes equation or equivalent chaotic system, we would love to compare with them.

---

### Official Review · Reviewer_NdmN · 2022-07-08

**Rating:** 7
**Confidence:** 4
**Soundness:** 3 good
**Presentation:** 4 excellent
**Contribution:** 3 good

**Summary:**

This article discusses robust estimation of chaotic dynamical systems, in particular their invariant statistics, using Markov neural operators (MNOs). In case of simple ODE systems, feedforward NN  instantiate the MNO, while Fourier operators are used for spatially extended (PDE) systems. The major innovation in the present work is the addition of a ‘dissipativity loss’ that enforces attracting behavior within a shell around the data, and thereby tends to enforce globally attracting properties. Essentially, the dissipativity loss tends to penalize larger flow vectors away from the data (why this works was not fully clear to me, however). This enables to learn highly challenging systems like the strongly turbulent Kolmogorov flow.

**Questions:**

1) If I understood correctly, a crucial premise of this approach is that there is indeed only one globally attracting limit set. This will be violated in most biological systems at least, for instance in molecular biology or neuroscience, as well as in many other complex dynamical systems of current interest, like climate models. For many real-world data from complex systems we would not know this in advance. Isn’t this a serious limitation of the approach?
Also, how does the vector field far from the attractor actually compare to that of the true system under this approach?

2) I felt that a lot of relevant literature was left aside (all the work by the Brunton group or Edward Ott’s group for instance, https://arxiv.org/abs/2201.05136, https://arxiv.org/abs/1904.02107, https://arxiv.org/abs/1710.07313, https://arxiv.org/abs/2110.07238, https://arxiv.org/abs/1712.09707, https://arxiv.org/abs/1910.03471, https://arxiv.org/abs/2110.05266, https://arxiv.org/abs/2005.10756, https://arxiv.org/abs/1707.01146, https://arxiv.org/abs/2207.02542, or the recent developments surrounding neural ODEs and PDEs, e.g. https://iopscience.iop.org/article/10.1088/1367-2630/abeb90). The authors are not the first to put the focus on invariant statistics; power spectra, Lyapunov spectra, or geometrical statistics have been discussed before in the context of reconstructing dynamical systems. Likewise, although the novelty about Theorem 1 may be its extension to spatio-temporal operators (function spaces), similar ideas I believe appeared before in the works of Funahashi and Nakamura or, more recently, by Raginsky and colleagues (https://proceedings.mlr.press/v120/hanson20a.html and references therein). Also, much stronger competitors are likely to be found among these previous methods than the plain LSTMs and GRUs used in Fig. 4 or the models used in Table 1. Many groups (like Brunton or Vlachas) have tested their approaches on similar systems like the Lorenz-63, Kuramoto-Sivashinsky or Navier-Stokes systems. These, in my mind, would have been the more relevant comparisons. This is especially relevant when the claim is made that other ML approaches fail to capture invariant statistics (lines 255-256).

3) If I understood correctly, the specific regularization solution is partly also motivated by the fact that iterating the MNO across many time steps (eq. 3) may easily lead to divergence. But this it seems to me is really a standard problem in RNN training, especially if the loss is only based on consecutive time steps. Specific algorithms to deal with this issue had been suggested before in the context of dynamical systems, e.g. https://arxiv.org/abs/2110.07238. Would be nice to know how these approaches differ in performance \& properties.

4) More methodological and algorithmic details in the main text would be appreciated. What is the loss function, apart from the novel regularization term? What precisely is the setup and parameterization of the forward NNs and the Fourier NOs, how were hyper-parameters of the architectures determined (especially $\lambda$ in the  dissipativity loss)?

Minor points:

- In my understanding RNNs also learn a Markov operator that is used as in eq. 3, what is the difference?

- Fig. 3 provides just single examples, likewise it appears Fig. 4. It would be good to see statistically more comprehensive evaluations with error bands on many different training runs. Also, it would be  good to see reconstructions of the actual attractors as well (the Lorenz is barely visible in Fig. 3).

- I found it somewhat difficult to follow parts of sect. 4.2 and Fig. 4, simply cos in my mind too few details were provided. What, for instance, is the ‘residual model’?


**Limitations:**

The major limitation I pointed out above was not discussed.


**Strengths And Weaknesses:**

Strengths:
In general I find this a sophisticated, concisely written and well presented contribution to the field from which I learned various things. This specific idea of enforcing dissipativity seems novel and useful to me.

Weaknesses:
... at the same time, for many systems this global form of dissipativity seems to be a quite strong and often too restrictive assumption, see my detailed Q below.
Much related lit. is not discussed, some methodological details are missing, for some results only single examples but no stats are presented.

---

> ### Author Response · Authors · 2022-08-02
> **Response to Reviewer NdmN**
>
>
> 1. Under our dissipativity assumptions (which many real-world systems satisfy, including most biological systems) in eq. 2, there exists a unique global attractor. We note that a global attractor *can contain* multiple attracting regions. The existence of a global attractor implies that the global attracting region is connected.
>
>      For systems with separate and distinct "sub-attractors" (e.g., systems with bifurcations) within the global attracting set, data can be collected separately across the entire global attractor so that the training set includes trajectories in the regions of interest.
>
>      Note that, although the global attractor is unique under the dissipativity assumption in our paper (eq. 2), the $\omega$-limit set (set of accumulation points of trajectories) of a single initial condition may differ across initial conditions [Stuart and Humphries, 1998; Section 2.8]. In this paper we concentrate on the ergodic setting where the same $\omega$-limit set is seen for almost all initial conditions with respect to the invariant measure. In future work it will be of interest to study non-ergodic problems in which different initial conditions lead to different $\omega$-limit sets; in particular it will be of interest to learn about the unstable manifolds which partition the state space into regions with differing $\omega$-limit sets. In future work it will also be of interest to study Hamiltonian, energy-conserving problems [Stuart and Humphries, 1998; Section 2.9].
>
>
> 2. Thank you for your time in collecting the relevant literature. We have added citations and highlighted key differences in our paper. Overall, we aim to systemtically study chaotic systems and enforce dissipaticity, which has not been studied in the previous works. We classify those works into four groups: (1) those learning continuous dynamics (e.g., SINDy), (2) techniques using reservoir computing, (3) technqiues using RNNs, and (4) methods for learning Koopman operators.
>
>     (1) **Learning continuous dynamics:** We purposefully avoid learning continuous dynamics (the time derivatives) for two reasons:
>     * To produce long-time predictions an integrator must still be used, which is computationally expensive.
>     * Learning continuous dynamics for PDEs is undesirable because it is an unbounded operator, which is more difficult to learn. [de Hoop et al., 2021] show, in the context of linear operators, that learning is easier (in terms of sample complexity) for compact and bounded operators than for unbounded operators. In contrast, our methods scale to infinite-dimensional PDE cases because we are learning the solution operator.
>
>     (2) **Reservoir computing:** Reservoir computing is similar to learning continuous dynamics. Usually the approximation power is limited: it requires a much smaller time-step (hence effectively learning continuous dynamics) and limits to a simpler model. To our knowledge, so far no reservoir computing model can predict the full velocity/vorticity field of a turbulent NS problem.
>
>     (3) **Recurrent neural networks:** The setting of our work is similar to the recurrent neural networks. However, prior works either fit the RNN on extremely short trajectories or simple, lower-dimensional system such as the Lorenz and KS equations. For Markovian chaotic systems, we show memory is not necessary to learn the solution operator.
>
>     (4) **Koopman operators:** In this line of work, the authors encode the dynamics into a latent space, and learn a linear operator on the latent space. However, even when the state space is finite dimensional, the Koopman approaches require approximation of a linear operator on a function space; when the state space is an infinite-dimensional approximation of this operator is particularly challenging.
>
> 3. Thanks for pointing out the reference! For chaotic systems, the Lyapunov exponents are greater than one which causes the system to diverge over a period. This is also the main conclusion of this paper: if the dynamics are chaotic, gradients of RNN will always explode. This is the main reason we do not use RNN/memory in our formulation.
> 4. Thanks for your comments. We have added a more precise description of our loss function (see eq. 5). We have also added ablation experiments for the hyperparameters of the dissipativity term in Appendix A.1, where we empirically show that the hyperparameters can be varied significantly without producing a large change in the performance of the model in terms of step-wise or dissipative error.

---

> > ### Author Response · Authors · 2022-08-02
> > **Response to Reviewer NdmN, 2**
> >
> > #### Minor points:
> > 1. RNN: Yes, the setting of our work is similar to recurrent neural networks. However, prior works either fit the RNN on extremely short trajectories or simple, lower-dimensional systems such as the Lorenz-63 and KS equations. For Markovian chaotic systems, we show memory is not necessary to learn the solution operator.
> > 2. Figure 3: We use Figure 3 to demonstrate the dissipativity of Lorenz-63 system. The statistical evaluation can be found in Figure 7 in the appendix. In the updated version, we zoom in the attractor and add a new sub-plot for the hard constraint case.
> > 3. Figure 4: By residual, we mean learning the time-derivatives. In the standard setting, we directly model the solution operator (flow map) $\phi:u(t) \mapsto u(t+1)$. The alternative setting is to learn $\frac{du}{dt} \approx (u(t+1) - u(t))*dt$. The two settings are equivalent. But when $dt$ is small, $u(t)$ and $u(t+1)$ are close, so the residual $u(t+1) - u(t)$ provides a stronger signal for learning.

---

> > ### Comment · Reviewer_NdmN · 2022-08-08
> > **Thanks for rebuttal and some further remarks**
> >
> > I thank the authors for their extensive revisions.
> > I wouldn’t agree with some of the points in the reply though: For instance, except for some strong simplifications of biological systems (e.g. Lotka-Volterra, SIR), most molecular, ecological or neuronal systems exhibit multiple attractors to my knowledge, and the same is true for many relevant complex physical systems (climate). Also, some of the lit. I collected in my initial review *does* study RNNs for long trajectories on high-dimensional systems. Thus, although the discussion of related lit. has been extended, I'm not sure the authors always appropriately acknowledge or do justice to this previous lit.
> >
> > These are not major points, I find the authors’ contribution valuable regardless, but something that the authors may want to acknowledge and discuss, perhaps with an outlook on how to deal with systems with multiple attractors.
> >
> > I'm still in favor of this work and certainly agree it’s very relevant for Neurips. But I’m not sure yet the revisions are fundamental enough to increase my rating. I feel some of the authors’ responses (1-3) slightly tend to navigate around my points rather than fully addressing them.

---

> > > ### Author Response · Authors · 2022-08-09
> > > **Response to Reviewer NdmN**
> > >
> > > Thanks for your feedback and your support. We are very glad to see you find our contribution valuable. The notion of dissipativity that we use includes problems, such as the Chaffee-Infante problem and the Brusselator, which possess multiple $\omega$-limit sets for individual points. However, because we also invoke **ergodicity**, such systems are excluded from our current study. We agree that the study of non-ergodic dissipative systems, and their basins of attraction, is an important direction for future study; we have added this outlook in Appendix C.
> > >
> > > The **multiple attractors** setting is also a very interesting direction. We would like to take the chance to clarify that our dissipativity formulation is applicable to multiple attractors. The definition of dissipativity we use in our paper defines the existence of an absorbing set $E$ for the system that consists of an open ball of radius $\sqrt{\alpha / \beta + \varepsilon}$, for any $\varepsilon > 0$ (see equation 2 in the paper).
> > >
> > > Whereas there can be multiple attracting subsets $E$ that may compete and provide differing dynamics within $E$, these systems are still dissipative with respect to $E$. In this paper, we seek to learn/enforce dynamics that remain globally dissipative with respect to $E$. As long as there is sufficient training data on the areas of interest within $E$ (e.g., multiple subsets of $E$ that form "sub-attractors"), MNO can be trained on these dynamics. Specifically, we will collect a dataset on each of the attractors, so the model can learn the dynamics on each of them.
> > >
> > > As for the RNNs for long trajectories in high-dimensional systems, we have cited this literature (~lines 58-59) and are grateful for the pointers. Nonetheless, as we clearly state (~lines 59-61) we believe our paper is the first paper to study the systematic imposition of dissipativity (according to our widely used definition) within machine learning.

---

> > > > ### Comment · Reviewer_NdmN · 2022-08-09
> > > > **final**
> > > >
> > > > The dissipativity aspect is indeed novel as far as I can see. What I meant is that limit behavior, invariant statistics and high-dimensional problems have been studied before in the RNN literature (some of the Vlachas papers, for instance, or https://arxiv.org/abs/2207.02542).
> > > >
> > > > I also got the point about the “sub-attractors”. The issue I have here is that practically, for an empirically observed complex physical or biological system, one usually wouldn’t know in advance whether it is multistable or not. If it is, imposing a dissipativity constraint would be the wrong thing to do, unless one is confident it’s restricted to a particular basin E.
> > > > Also, the dissipativity constraint induces a strong bias in the vector field farther away from the attractor and potentially close to the border of E for a multiple attractor system. This is another non-trivial issue I think, it may destroy certain topological properties of the system.
> > > >
> > > > Nevertheless I feel this is a worthwhile direction to consider, and either way a technically superb contribution. I also appreciate the authors widened their discussion of related work.
> > > > I therefore decided to raise my score further.

---

> > > > > ### Comment · Reviewer_NdmN · 2022-08-09
> > > > > **(cont'd)**
> > > > >
> > > > > ... I'd still be interested to hear the authors' opinion on the caveats I mentioned in my last reply!

---

> > > > > > ### Author Response · Authors · 2022-08-09
> > > > > > **Response to Reviewer NdmN**
> > > > > >
> > > > > > We really appreciate your support and encouragement! Your feedback helped us improve the paper. We agree with the reviewer that the major assumption is dissipativity. Such assumption does not hold for energy-conservative hamiltonian systems such as the solar systems. In this case, it will be wrong to impose a dissipativity constraint. Besides, in the multiple-attractors scenarios, it can be non-trivial to obtain a representative dataset for each of the sub-attractor, which requires gathering the train data actively or using importance sampling techniques.

---

### Official Review · Reviewer_3spS · 2022-07-11

**Rating:** 7
**Confidence:** 4
**Soundness:** 2 fair
**Presentation:** 3 good
**Contribution:** 2 fair

**Summary:**

This paper proposes a machine learning framework, named Markov neural operator (MNO), to learn the Markov operator of dissipative chaotic systems. Different from the conventional neural networks, they use Sobolev norms in operator learning and add dissipativity losses to ensure dissipativity. They experimentally show that the MNO can accurately approximate the global attractor and estimate various statistics of the invariant measure for dissipative chaotic systems. Besides, they provide a theoretical guarantee that this model is rich enough to approximate many chaotic dynamical systems for an arbitrarily long period.

**Questions:**

1. How the choice of the inner and outer radii of the shell of the dissipativity losses influences the performance of the MNO? Whether the choice significantly affects the MNO’s step-wise error and the estimation of the statistical properties of the attractor? How to choose the radii based on the training data?
2. How to balance the influences of the Sobolev and the dissipativity losses? Is there a hyperparameter to adjust their proportion in the loss function?
3. Does the choice of lambda significantly influence the performance of the MNO? How to choose the optimal lambda according to the training data?
4. Is it possible to provide a method to judge whether the MNO can be directly applied to a certain target system?


**Limitations:**

The MNO can be used only when the dynamical system is Markovian, but actually we do not know whether that assumption holds in a model-free task. This paper lacks a method to judge whether the MNO can be directly applied to the certain target system.

**Strengths And Weaknesses:**

This is a novel method to approximate the global attractor and estimate the invariant statistics of dissipative chaotic systems. They pioneered the combination of the Sobolev losses and the dissipativity losses, which improves the performance for preserving the invariant statistics. The dissipativity losses they proposed can enforce the dynamics close to the attractor, which improves the robustness against large perturbations. They also investigate the effect of time steps and different Sobolev losses in operator learning, which makes this work more systematic. However, there are still some confusing details. The choice of the hypermeter lambda and the inner and outer radii of the shell in the dissipativity losses remains to be discussed. There is no description of the proportion of the Sobolev and the dissipativity losses in the loss function. In addition, sometimes we do not know whether the target system is Markovian in a model-free task. As a result, this paper lacks a method to judge whether the MNO can be directly applied to a certain target system.

---

> ### Author Response · Authors · 2022-08-02
> **Response to Reviewer 3spS**
>
>
> 1. Thank you for your questions. To clarify, we have empirically found that our trained models appear robust to variations in the shell radius of the enforced dissipativity. We have added ablation experiments in Appendix A.1 to corroborate our claims. In general, we believe it is ideal to select an inner radius that is solely dependent on the training set and on $\lambda$. We choose our inner radius to be $r_i = \sup \frac{\|x\|}{\lambda}$ over all $x$ in the training set. Note that if $\|x\| < r_i$, then $\lambda x$ would lie inside the outer-most point of the attractor (assuming the training set adequately captures the attractor). We have found that the outer radius $r_o$ of the shell typically does not matter significantly because the models tend to learn dissipative dynamics outside of $r_o$ as well.
> 2. We appreciate your questions on balancing the Sobolev loss and dissipativity regularization. To clarify this point, we have clearly defined our Sobolev norm step-wise loss term (eqs. 13-14) and the overall loss function used to train our MNO (eq. 5).
> 3. We have added further ablation experiments in Appendix A.1 to address this point. We have found that the step-wise error and dissipativity error of our trained models is not highly sensitive to the choice of $\lambda$ and $\alpha$ (the weight between the Sobolev norm and dissipativity loss terms). As for the choice of $\lambda$, this may be application-dependent. For certain systems where returning to the attractor quickly is important, larger values of $\lambda$ may be used. Our results suggest that reasonable values of $\lambda$ should not significantly increase the error of the trained model.
> 4. We acknowledge that we are only focusing on learning Markovian systems in this work. However, there are many systems of practical scientific and engineering value (e.g., Navier-Stokes as presented in our paper) for which a priori we know there exists a Markovian solution operator, on which our techniques could be applied.

---

### Author Response · Authors · 2022-08-02
**General responses**

We thank the reviewers for their helpful comments and suggestions. We have updated the paper and used color code to show the changes. For better presentation, please open the link to view the full response with figures: https://hackmd.io/@anonymous-author/B1NR1ePaq

Overall, we make three major updates:
### Hard-constraint: Enforcing dissipativity via post-processing:

![](https://i.imgur.com/jbiCVme.jpg)

While the previous dissipativity regularization encourages a stable map in practice, it does not guarantee that dissipativity is enforced. For an additional safeguard against model instability, we enforce a hard *dissipativity constraint* far from the attractor and from the shell where $\nu$ is supported. This allows for provably dissipative dynamics on a ball sufficiently far from the attractor.

Specifically, we post-process the model: whenever the dynamic moves out of an a priori defined stable region, we switch to the second model $\Psi$ that pushes the dynamic back. The new model combines the learned model $\hat S_h$ and the safety model $\Psi$, via a threshold function $\rho$:
\begin{equation}
    \hat S_h' (u) = \rho(\|u\|) \hat S_h + (1 - \rho(\| u\|)) \Psi(u),
\end{equation}
where $\Psi$ is some dissipative map and $\rho$ is a partition of unity. For simplicity we define
\begin{equation}
    \Psi(u) = \lambda u \hspace{4em} \rho(\|u\|) = \frac{1}{1 + e^{\beta(\|u\| - \alpha)}},
\end{equation}
where $\alpha$ is the effective transition radius between $\hat S_h$ and $\Psi$ and $\beta$ controls the transition rate. Note that this choice of $\Psi$ is consistent with the regularization term in the loss (eq. 5).

### Kolmogorov flow with $Re=5000$:

![](https://i.imgur.com/DTutGuU.jpg)

The proposed dissipative MNO model can learn stable dynamics with a high Reynolds number. As shown in the figure above, the model correctly captures the build up of the turbulence. The first column corresponds to the initial condition and is sampled from a Gaussian random field. Around $t=10$ to $t=20$, we can see the energy injected from the source term $\sin{(4y)}$ (see equation 10). Around $t=10$ to $t=20$, the energy dissipates into the lower frequencies, and the spectrum converges to the ground-truth Kolmogorov energy cascade rate of $k^{-5/3}$. The learned model is stable with the dissipative regularization. Even with a $O(10^4)$  perturbation, the dynamic quickly returns to the attractor of the system. In contrast, the model without dissipative regularization will stay outside the attractor with a much smaller  $O(10)$ perturbation.

### Ablation study on the hyperparameters for the dissipativity regularization:

![](https://i.imgur.com/DnJ9LmK.png)


Ablation experiments show the effect of varying dissipativity hyperparameters. Error rates are given as relative $L^2$ error. Per-step error, per-second error, and dissipativity error are defined as in Table 1 in the paper. Unaltered hyperparameters are held constant at default values of a radius of $40$, $\alpha=1$, and $\lambda = 0.5$.

---

### Author Response · Authors · 2022-08-07
**Looking forward to your responses**

Dear reviewers,

There is a gentle reminder that the discussion period is going to end soon. Please feel free to let us know if our responses help to clarify your questions and concerns, so we can make further explanations and add new experiments if needed.

We really appreciate the reviewers’ actionable feedbacks, and we made great efforts to address with additional numerical studies on hard-constraint dissipativity, the ablation study for the regularization loss, and the turbulent Re=5000 system. We hope that the Reviewer will take these new results into account and increase our score if we have adequately addressed the main concerns.

Thank you,
Authors

---

### Meta-Review · Area_Chair_WK4B · 2022-08-25

**Recommendation:** Accept
**Confidence:** Certain

**Metareview:**

This paper proposes a neural network-based approach to estimate the Markov operator of dissipative chaotic systems. It introduces a novel combination of Sobolev and dissipativity losses. While the reviewers had initial concerns about clarity, assumption and application condition, and the choice of learning Markov operator versus modelling continuous dynamics, the author-reviewer discussion addressed most concerns, and all reviewers agree this work exceeds the bar for publication.

I would encourage the authors to take into consideration the remaining concerns from the reviewers, incorporate key conclusions of the discussions and the limitation of the work in their final version.

**Award:**

No

---

### Decision · Program_Chairs · 2022-09-14

Accept